# SIGHT: Single-Image Conditioned Generation of Hand Trajectories for 3D Hand-Object Interaction

## Abstract

We introduce a novel task of generating realistic and diverse 3D hand trajectories given a single image of an object, which could be involved in a hand-object interaction scene or pictured by itself. When humans grasp an object, appropriate trajectories naturally form in our minds to use it for specific tasks. Hand-object interaction trajectory priors can greatly benefit applications in robotics, embodied AI, augmented reality and related fields. However, synthesizing realistic and appropriate hand trajectories given a single object or hand-object interaction image is a highly ambiguous task, requiring to correctly identify the object of interest and possibly even the correct interaction among many possible alternatives. To tackle this challenging problem, we propose the SIGHT-Fusion system, consisting of a curated pipeline for extracting visual features of hand-object interaction details from egocentric videos involving object manipulation, and a diffusion-based conditional motion generation model processing the extracted features. We train our method given video data with corresponding hand trajectory annotations, without supervision in the form of action labels. For the evaluation, we establish benchmarks utilizing the first-person FPHAB and HOI4D datasets, testing our method against various baselines and using multiple metrics. We also introduce task simulators for executing the generated hand trajectories and reporting task success rates as an additional metric. Experiments show that our method generates more appropriate and diverse hand trajectories than baselines and presents promising generalization capability on unseen objects. The accuracy of the generated hand trajectories is confirmed in a physics simulation setting, showcasing the authenticity of the created sequences and their applicability in downstream uses.

## 1 Introduction

As our hand grasps an object, we immediately plan out potential maneuvers to manipulate it for our intentions. Consider pouring some juice from a bottle into a cup – it can be as straightforward as rotating your wrist. At a granular level, it requires a continuous adjustment of the hand translation and orientation to transfer exactly the desired amount of liquid into the target receptacle. Humans' hand motion planning systems are remarkably robust in adapting to unseen objects, emulating movements observed from others, and devising paths from visual cues alone. Robotic agents and AI systems alike could benefit immensely from a similar ability to synthesize 3D hand trajectories from an image depicting a hand-object interaction scenario, be it to anticipate human behavior, generate realistic animations, or interact with the physical world. Following this realization, we wish to investigate the generation of high-quality hand trajectories from visual representations of objects.

In this paper, we propose the new task of *Single-Image Conditioned Generation of Hand Trajectories* (SIGHT). Given a single first-person image showing a hand interacting with an object, our goal is to generate plausible and diverse 3D hand trajectories that meaningfully complete the action initiated in the image. Additionally, we explore the generation of appropriate interaction trajectories from standalone images of previously unseen objects without a hand manipulating the object.

---

[0]We will release our code, benchmark, and results to the public.

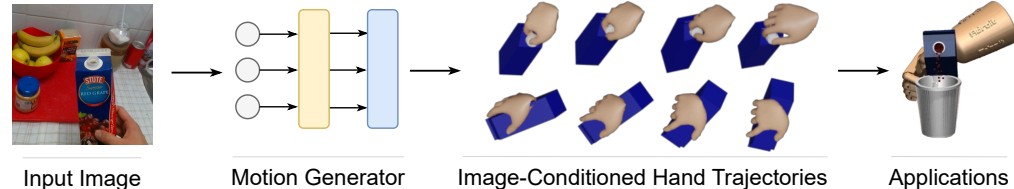

| Input Image | Motion Generator | Image-Conditioned Hand Trajectories | Applications |

Figure 1: **Task description.** Given an input image showing an object, either being interacted with by a hand or pictured by itself, we propose the novel SIGHT task, requiring the generation of task-adequate, diverse, and realistic hand sequences mapping out possible trajectories of the hand when interacting with the object as shown. The task has applications in robotics, character animation, and human intent prediction, among others.

Previous studies on hand-object interactions concentrate on detecting and segmenting (Darkhalil et al. (2022); Zhang et al. (2022a)) pairs of hands and objects interacting within images, or aim to reconstruct the 3D models of objects and the hands interacting with them from images (Hasson et al. (2019); Ye et al. (2022)) or videos (Fan et al. (2024); Ye et al. (2023a)), but do not generate trajectories. Furthermore, the field of human motion generation has hitherto focused on whole-body motion synthesis (Guo et al. (2022; 2020b); Tevet et al. (2022)), with little attention paid to synthesizing realistic, task-appropriate and interactive hand motions. The conditional information in previous whole-body motion synthesis work is restricted to either action labels or textual descriptions of motions, inconveniently requiring manual text annotations or action descriptions alongside the motion data. In contrast, our main focus is on generating dynamic 3D hand trajectories from a single static image. We utilize an underexplored and more easily affordable form of conditioning motion generation models to generate motions for an important yet neglected facet of human motion. Our aim is to reduce the amount of conditional information provided, restricting it solely to the static hand-object interaction or standalone object portrayed in the input image. Diverging from prior hand-object interaction tasks, the new proposed task, SIGHT, has a more generic but challenging setup, considering the multitude of possible hand-object interactions and the minimal guidance information provided.

SIGHT comes with several challenges. When generating trajectories from first-person hand-object interaction images, the initial step in addressing this task is to detect the interacting hand and object, as images captured from a real-world first-person perspective often exhibit significant clutter in the background. Moreover, generating future hand trajectories from images capturing static moments of interaction involves disambiguating the intended action, with objects often having multiple possible uses corresponding to different hand trajectories. The points of contact between the hand and the object play a key role in differentiating between similar actions, although hand and object often occlude each other in the images. Further, synthesizing sequences of hand trajectories necessitates the generation of smooth, natural and anatomically plausible motions that lead to the successful completion of the initiated actions for contacted objects. Finally, an interaction trajectory generation system integrated into the real world must operate robustly even when presented with previously unseen objects, complicating the inferral of object-appropriate trajectories.

We present SIGHT-Fusion to address these challenges. Our key insight is in the meaningful identification of the contacted object in presence of distracting information in the input image's background so as to maximally disambiguate the conditioning of the motion generator. Further, we find that egocentric videos of people interacting with diverse objects can serve as an effective source of training data for reasoning about interaction trajectories fitting previously unseen object instances. The effective use of visual hand-object interaction features enables our model to infer the underlying intentions and generate realistic hand trajectories for the intended interactions depicted in a single image. More specifically, we propose SIGHT-Fusion, a conditional motion generation diffusion model that learns to generate realistic and diverse 3D hand trajectories. The extracted hand-object interaction features are used as the conditional input. Through training, SIGHT-Fusion learns the distribution of possible hand movements given a static interaction moment and acquires the ability to distinguish between potential plausible motion sequences by leveraging fine-grained details contained in the contact regions. Using a general-purpose vision foundation models for the extraction of object features, SIGHT-Fusion effectively generalize to object instances not seen during training.

We set up comprehensive baselines and metrics utilizing the FPHAB (Garcia-Hernando et al. (2018)) and HOI4D (Liu et al. (2022)) datasets for the newly proposed SIGHT task. Extensive experiments show that SIGHT-Fusion is able to generate more natural and diverse hand trajectories than baselines. We also observe promising potentials for our learned system to generalize to unseen object instances. We further introduce a task-oriented metric to execute the generated hand trajectories in physical simulation and evaluate if they could lead to successful task execution. Results show that our method is able to produce trajectories that can accomplish downstream tasks successfully. Lastly, we empirically validate system designs including the usefulness of various feature types in improving the quality of our result.

In summary, the contributions in this paper are as follows: 1) We introduce a novel task of generating 3D hand trajectories given a single image of an object, in the two settings of either being depicted by itself or being interacted with by a human hand. 2) We set up comprehensive benchmarks for the new task, including various baselines and metrics. We further propose a task success rate metric for evaluation using physical simulation. 3) We develop SIGHT-Fusion to tackle the proposed problem with a novel pipeline to extract hand-object interaction features from the single image input and a conditional diffusion-based hand trajectory generative model. 4) Experiments show superior performance compared with baselines and ablated versions of our system.

## 2 RELATED WORK

The field of learning hand-object interactions, particularly in first-person videos, has seen growing interest. We briefly discuss existing methods for hand-object interaction segmentation, 3D hand reconstruction, and motion generation.

**Hand-Object interaction segmentation.** Advancing the domain of interaction detection, Shan et al. (Shan et al. (2020)) introduce a method based on Faster-RCNN (Ren et al. (2015)) to detect hands and objects from RGB images, utilizing RoI features for predicting hand sides, bounding boxes, and contact states. EgoHOS (Zhang et al. (2022a)), adopting a similar framework, enhances the methodology by training on diverse first-person datasets and adding the capability to predict the contact boundary between hands and objects. In a different approach, COHESIV (Shan et al. (2021)) generates detailed pixel-wise features, enabling the segmentation of images into classes of people, objects, and backgrounds. This segmentation process is augmented by a user-defined query point on the visible hand, utilizing optical flow and regressed hand poses to improve accuracy.

**Whole-Body motion generation.** Recent work in motion generation has relied heavily on Variational Auto Encoders (VAE) and Diffusion models. Studies have explored conditional generation using text (Karunratanakul et al. (2023); Kim et al. (2023); Tevet et al. (2023); Zhang et al. (2022b)), audio (Yi et al. (2023)), both (Dabral et al. (2023); Zhou & Wang (2022)) and categorical actions (Zhao et al. (2023)). Tevet et al. (Tevet et al. (2023)) expand upon previous work (Kim et al. (2023); Zhang et al. (2022b)) by introducing a classifier-free diffusion model MDM for text-driven full-body human motion generation. Other works such as (Diomataris et al. (2024)) also generate full-body human motions using c-VAEs. GMD (Karunratanakul et al. (2023)) further enhances it by adding spatial constraints. Most works such as (Petrovich et al. (2022b)) use VAE's to generate diverse SMPL body shapes from texts, while others (Yi et al. (2023); Zhang et al. (2023)) use a Vector Quantised VAE (VQ-VAE) to generate human poses from text and speech respectively. A hybrid approach combining VAEs and diffusion models by (Chen et al. (2023)) de-noise conditional latent vectors and decode them to produce human body poses.

**Hand motion generation.** A novel two-step method is introduced in Ye et al. (2023b) for synthesizing hand-object interaction images from a single RGB image of an object using diffusion models. This method, however, generates static snapshots rather than a continuous motion. Other works such as HMP (Duran et al. (2023)) try to generate 3d hand poses even if the hand is occluded. All of these approaches have the disadvantage that they generate snapshots rather than full motions which is what has been done for full body movements. Bao et al. (2023) tries to predict hand movement for a VR/AR setting however it only predicts the general motion and not the precise movement of individual joints. AI often has problems with generating accurate and precise hands, which is often used as a telltale sign to differentiate actual images from generated ones.

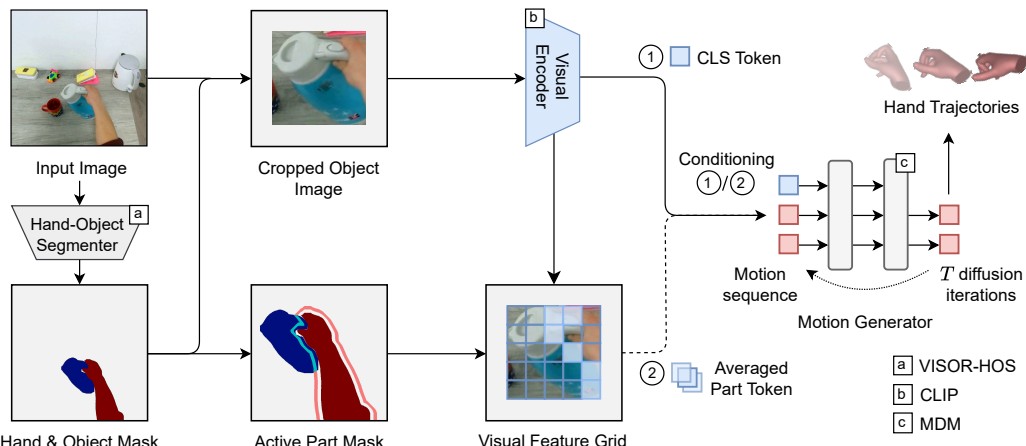

Figure 2: **Method description.** Given the input image, we first use a hand-object interaction detection system to locate the interacted object and hand, after which we extract object and part features from the image. The features are forwarded to a diffusion-based motion generator, which produces task-appropriate and realistic hand trajectories.

**Motion in simulator.** Manipulation tasks using hands inside a physical environment is a challenging as integrating forces such as gravity and current velocity of an object can render seemingly good trajectories bad. When considering the task of pouring juice into a glass moving the juice bottle with to much speed quickly leads to liquids spilling in all directions. Work in this area uses methods such as RL learning (Rajeswaran et al. (2018)) to generate 3D realistic motions. Some prior works explored using physical simulators to evaluate the stability of synthesized grasps (Yang et al. (2021); Wang et al. (2024)), considering only a static scenario. However, to the best of our knowledge, there have not been any papers where the authors try to evaluate the physical realism of their generated trajectory with a physical simulator.

## 3 METHOD

In this section, we start by formulating the SIGHT task (Section 3.1). We then detail our feature extraction method (Section 3.2), which is used to obtain conditioning inputs for training and synthesizing hand trajectories with our motion generation model SIGHT-Fusion. Finally, we describe our model design for generating accurate, diverse, and realistic hand trajectories (Section 3.3). Our feature extraction method and model are visualized in Figure 2.

### 3.1 TASK DEFINITION

We consider two variations of our task: in our first setup, the input $\mathcal{I}$ to the hand motion generator $\mathcal{M}$ is a first-person image showing a hand enacting a certain action on an interacted object, such as *opening* or *pouring out* a bottle. In our second setup, $\mathcal{I}$ consists of a close-up image showing an object without an interacting hand. For both cases, the expected output is a sequence of hand poses $\mathcal{H}_{1:f} = \mathcal{H}_1, \ldots, \mathcal{H}_f$, where $f$ is a predefined number of frames. Each $\mathcal{H}_i$ corresponds to a representation of a 3D human hand for a single timestep. Our work represents the $\mathcal{H}_i$ using SMPL-X (Pavlakos et al. (2019)) and reasons about the right hand, in concordance with the almost exclusively right-handed object interactions seen in most publically available video datasets.

### 3.2 EXTRACTING ACTION-INFORMATIVE FEATURES

Given an input image, the SIGHT task requires extracting highly informative visual features to guide the motion synthesis. When generating hand trajectories from images depicting hand-object interaction scenarios, correctly identifying the object of interest in the input frame presents a challenge due to background clutter and occlusion caused by the hand. The synthesis of a correct motion is

even more difficult when generating trajectories from images of standalone objects not including manipulating hands, as no interaction cues are given.

**Object features.** We first use VISOR-HOS (Darkhalil et al. (2022)) to detect interacted objects $\mathcal{O}$ in the input images $\mathcal{I}$, and then crop to the bounding box of $\mathcal{O}$ before extracting visual features. See the Supp. Mat. for details about the training of VISOR-HOS. The visual object features are obtained using CLIP (Radford et al. (2021)) to leverage the generalizability of foundation model features, helping us reason about objects not seen during training.

**Part features.** Furthermore, we investigate providing the motion generator with additional conditioning information in the form of features local to the contact point between the hand with the object. We extract features from high-level CLIP 2D feature grids and use them as part features. Our hypothesis is that part features help to disambiguate different actions that are associated to different object parts. We begin by dilating the HOS hand mask and intersecting the dilated mask with the HOS object mask. The resulting mask is then cropped to the region of $\mathcal{O}$ and resized to match the dimension of the 2D CLIP feature grid. An average feature vector is computed by taking the mean of all grid patches within the resized mask region. This vector is linearly projected to the input dimension of our Transformer (Vaswani et al. (2017))-based motion generator and prepended to the input motion sequence.

### 3.3 LEARNING TO GENERATE HAND TRAJECTORIES

Denoising diffusion models, originally proposed for image synthesis (Rombach et al. (2022); Saharia et al. (2022)), have been successfully adapted to whole-body human motion generation (Tevet et al. (2022); Karunratanakul et al. (2023); Yuan et al. (2023)). These models surpass traditional methods based on GANs (Barsoum et al. (2018); Guo et al. (2020b)) and VAEs (Petrovich et al. (2021; 2022a)) by delivering superior quality and diversity in generated motions. Given the shared requirement of reasoning about temporal sequences of joints, we adapt the state-of-the-art Human Motion Diffusion Model (MDM) (Tevet et al. (2022)) for the SIGHT task of 3D hand trajectory generation. Originally, MDM synthesizes whole-body human motions from text inputs or learned embeddings encoding action labels. The motion synthesis process for a sequence of $N$ frames, represented by $J$ joints with $R$ features each, starts with a randomly sampled latent code $x_T \sim \mathcal{N}(0, I)$ where $x_i \in \mathcal{R}^{N \times J \times R} \, \forall i \in \{1, ..., N\}$, and uses a transformer $\mathcal{G}$ to iteratively diffuse the final motion $x_0$ in $T$ steps.

We employ a Transformer encoder model to iteratively diffuse the synthesized motion, starting from Gaussian noise. During each diffusion iteration, the Transformer receives as input the conditioning information contained in the initial tokens, followed by tokens representing the motion sequence from the previous diffusion step, or Gaussian noise for the initial step (see Figure 2). Our work does not consider the prediction of manipulation trajectories for multiple objects being manipulated by the actor simultaneously, as this is a highly specific setup with little publicly available data.

We repurpose MDM for hand trajectory generation by replacing the original whole-body kinematic tree with the representation of the right hand used by OpenPose (Cao et al. (2017)) and SMPL-X (Pavlakos et al. (2019)), yielding $J = 17$ joints. We also replace the original textual encoder used for conditioning inputs with CLIP's visual encoder. We maintain the 6-dimensional pose representation used by MDM to encode motions, i.e. $\mathcal{R} = 6$, as suggested by (Zhou et al. (2019)).

To train our motion generator, we adapt the simple position and velocity losses from Tevet et al. (2022) while dropping the foot contact loss. Specifically, let $c$ be the conditioning information, $FK$ be a forward kinematics function reconstructing 3D hand joints from 6D pose representations, $r$ and $\hat{r}$ be the original and reconstructed 6D pose representations, and $x_t^i$ be the full pose representation at diffusion step $t \in \{1, ..., T\}$ for frame $i \in \{1, ..., N\}$. During every diffusion step $t$, we concatenate the input conditioning information $c$ with $x_t$ to form the input of $\mathcal{G}$, which is trained to predict the final motion sequence from this condition-augmented intermediate representation. Our optimized loss term $\mathcal{L}$ hence consists of:

$$\mathcal{L} = \mathcal{L}_{simple} + \lambda_{pos}\mathcal{L}_{pos} + \lambda_{velocity}\mathcal{L}_{velocity}, \tag{1}$$

where

$$\mathcal{L}_{simple} = \mathbb{E}_{x_0 \sim q(x_0|c), t \sim [1,T]} \big[ \|x_0 - \mathcal{G}(x_t, t, c)\|_2^2 \big],$$

$$\mathcal{L}_{pos} = \|FK(r) - FK(\hat{r})\|_2^2,$$

$$\mathcal{L}_{velocity} = \frac{1}{N-1} \sum_{i=1}^{N-1} \|(x_0^{i+1} - x_0^i) - (\hat{x}_0^{i+1} - \hat{x}_0^i)\|_2^2.$$

## 4 EXPERIMENTS

We demonstrate the effectiveness and generality of our proposed SIGHT-Fusion on a diverse set of in-the-wild egocentric videos. We first introduce the experimental setup in Section 4.1. Next, we show the quantitative and qualitative evaluation results of the generated hand trajectories in Section 4.2, and highlight the model's ability to generalize to unseen objects in Section 4.4. Lastly, we evaluate the generated hand trajectories in a physics simulator, demonstrating the physical realism of the generated motion sequences (Section 4.6).

### 4.1 DATASETS

To establish a comprehensive evaluation for our newly introduced task, we adapt two first-person video datasets for our benchmark. We also introduce new dataset splits to test adaptability to unseen scene backgrounds and new object instances.

**FPHAB** (Garcia-Hernando et al. (2018))    The FPHAB dataset contains first-person videos capturing 45 activities (defined as action-object pairs, e.g. *stir cup*), involving 26 objects. Each activity is recorded multiple times by six different subjects, all using their right hands. Motion-captured hand poses are provided together with the video data. Following the human motion generation literature, we first merge all actions of the same verb, disregarding the corresponding objects (to be elaborated in Sec. 4.2). We further eliminate videos without any object manipulation, such as those where a simple handshake is performed. We note that most of the frames in FPHAB contain hand-object interactions and pure object frames without occlusions are rare to find.

*Preprocessing.* Notably, the hands of all subjects in FPHAB exhibit significant occlusion due to motion capture markers, which leads to a performance deterioration of the off-the-shelf hand-object interaction detector used in our pipeline (Darkhalil et al. (2022)). To address this issue, we inpaint away these markers using a video inpainting method (Li et al. (2022)) prior to processing the dataset. Please refer to Supp. Mat. for detailed evaluation and visualizations.

*Data split.* We use the *cross-subject split* proposed in the original work, with subjects 1, 3 and 4 used for training, and the three others used for testing. As all actions, object categories and object instances appear in both the training and test sets, this original FPHAB split poses a simple test verifying the generalizability of models to novel views of object instances and actions already seen in the training set. However, the high number of objects and the association of multiple actions with the same object category make inferring the correct action to synthesize appropriate hand trajectories particularly challenging for this dataset.

**HOI4D** (Liu et al. (2022))    The HOI4D dataset consists of first-person videos of hands interacting with everyday objects from 16 categories. The number of object instances varies within each category, ranging from 31 to 47, and the action tasks associated with these objects vary between 2 to 6 per category. Altogether, the dataset defines 31 action tasks. As many tasks are highly similar to each other or only involve a displacement of the object, we merge/rename certain tasks and drop others to extract 13 *actions* from the original 31 tasks. Details of this grouping are provided in the Supp. Mat. As no public data split is available for HOI4D, we define our own splits with the aim of testing several dimensions of the generalizability for methods addressing the SIGHT task.

*Instance split.* For each object category and each action, there exist videos of at least 2 different object instances in HOI4D. This lends itself well to grouping the videos corresponding to different object instances of the same category and showing the same action being enacted. We then split each group into one subgroup with instances to use only for the training set, and another subgroup with instances to use only for the test set. Our proposed *instance split* thus allows a meaningful evaluation of cross-instance action knowledge transfer. Simultaneously, more advanced reasoning is required for successful cross-instance transfer, as object instances in the test set may look different

from those seen during training. Unlike in FPHAB, the videos in HOI4D do not start with the manipulated object already in hand. We thus construct the test frames for the instance split by cropping the first video frame to the yet ungrasped object of interest. Altogether, the instance split allows us to evaluate the setting of generating 3D hand trajectories for an unseen object, without interaction cues provided by the presence of a hand in the image. A detailed description of the proposed instance split is provided in the Supp. Mat.

*Location split.* We further wish to investigate generalization ability for hand-object interaction scenes explicitly featuring hands in view. Since the HOI4D videos feature objects recorded at distinctive locations associated with specific actions, the background becomes more visible in the input image when expanding to a larger region to include the hand into the input image. We hence design the *location* split by dividing the HOI4D videos into training and test sets according to their recording location, i.e. the environment. Doing so reduces the possibility of data leakage through the image's background. The location split hence provides an evaluation for the scenario of encountering both seen and novel objects in environments that are unseen during training.

**In-contact frame selection.** When constructing the test sets of our datasets in scenarios where the object and hand are in contact, we use VISOR (Darkhalil et al. (2022)) to detect the frame where the actor's hand first contacts any object in each video. We then manually verify that the actor is indeed contacting the *desired* object. If necessary, we select a later frame that shows actual contact with the desired object. This manual adaptation is needed for only a few videos and is done to ensure the reliability of the test data used for evaluating methods.

## 4.2 Evaluation of Generated Hand Trajectories

**Evaluation metrics.** Following the combination of metrics that are commonly used in the human motion generation literature (Guo et al. (2020b); Tevet et al. (2022)), we measure a method's success on our task based on the *accuracy*, *diversity*, and *fidelity to ground-truth* of its generated trajectories.

- **Accuracy** (ACC): Given that each generated hand trajectory is supposed to depict a specific action, we use the accuracy of an action classifier working on hand trajectories to measure the accuracy of the trajectories produced by a motion generation method $\mathcal{M}$. This formulation encourages the development of methods adept at matching their output trajectories to the hand-object interaction in the image given visual information alone, with no explicit knowledge of the desired output action.

- **Diversity** (DIV): The diversity of $M$'s outputs is calculated through the Fréchet Inception distances (FIDs) (Heusel et al. (2017)) between action classifier features extracted from generated (gen.) and ground-truth (GT) motion groups. The diversity of the generated trajectories corresponds to the FID of one half of a trajectory group to the other half. It is desirable in moderation: too little diversity corresponds to a method simply (re-)producing a few learned trajectories, while too much results in the hand trajectories becoming erratic. Hence, a diversity score close to that of the GT trajectories is desirable.

- **Fidelity** (FID): The fidelity of hand trajectories generated by $\mathcal{M}$ is calculated through the Fréchet Inception distances, similar to the diversity metric. Lower FIDs between generated and ground-truth sequences translate to a motion generation method producing trajectories more similar in distribution to real reference trajectories.

**Implementation details.** For each run, consistent with the human motion generation literature (Guo et al. (2022; 2020a); Tevet et al. (2022)), we select the checkpoint achieving the lowest FID metric on the test set, so as to produce the trajectories most similar to the test set trajectories. Hyperparameters are provided in Supp. Mat.

**Baselines.** We compare our method against several strong baselines. As the proposed SIGHT task is novel and no existing approach is suitable for solving the task, we adapt state-of-the-art baselines from the whole-body motion generation literature. We evaluate both image-conditioned and text-conditioned setups. The image conditioning features are produced as described in Sec. 3.2. The text conditioning is obtained by prompting LLaVa (Liu et al. (2023)), a state-of-the-art visual question answering model, with the question *"What is the hand in the image doing?"* given the input image

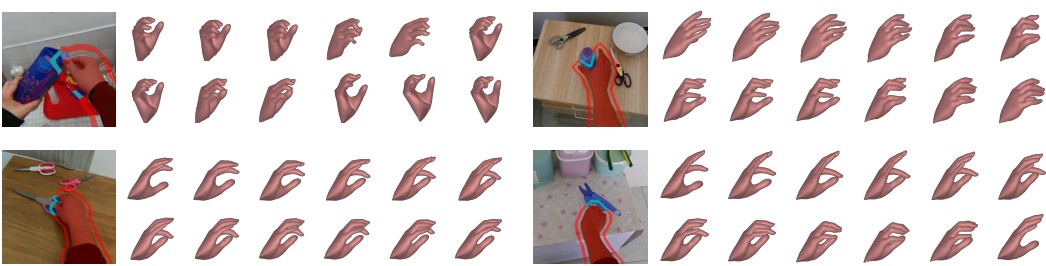

Figure 3: **Examples of generated trajectories.** We visualize hand trajectories generated by our $\mathcal{P}$ method when conditioned on the given input images. We further visualize the inferred object regions (blue), as well as part regions (cyan) from which part features are calculated.

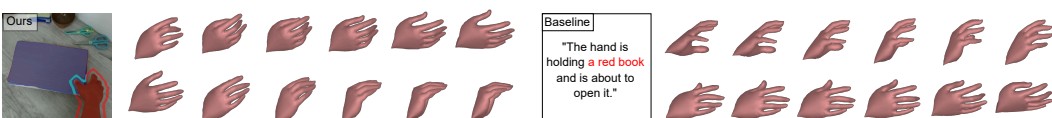

Figure 4: **Comparison with text-based baseline.** We visualize the input image, the inferred object (blue) and part (cyan) regions, the corresponding trajectory generated by our $\mathcal{P}$ model in the upper row, as well as the VLM prompt output and the corresponding generated trajectory in the lower row. Contrary to the baseline, Our method directly leverages visual features and produces an adequate trajectory by avoiding translation errors from the VLM (see Sec. 4.2).

cropped to the region of the manipulating hand and object. This text extraction helps mimic setups where we do not have access to ground-truth action annotations, as otherwise the synthesis task becomes trivial.

To evaluate the usefulness of our object and part features, we compare against an image-based version of MDM processing uncropped frames, which we call *MDM-I*. We further compare against the text-based version of MDM (*MDM-T*) and the text-based work of Guo et al. (2022), herein called *T2M-T*. See Supp. Mat. for details.

### 4.3 COMPARISON WITH STATE-OF-THE-ART

The evaluation results in Table 1 validate our hypothesis that conditioning on the interacted object leads to more appropriate trajectories, as evidenced by the increase in accuracy when visual features are used as input. The VLM-based image-to-text translation process is noisy and results in an inferior accuracy of the text-based methods, as also illustrated in the example comparison between T2M-T and our method in Fig. 4. This highlights the usefulness of visual input features for motion synthesis.

Compared to the evaluated baselines, our method is able to achieve significantly better performance in both accuracy and FID scores on the FPHAB dataset and accuracy on the HOI4D location split. We further achieve compatible results for the diversity on the HOI4D location split. We observe that image features lead to better trajectory generation evaluated on both accuracy and FID. Compared to object features, part features lead to better accuracy on both datasets. Examples of trajectories generated by our models are visualized in Figure 3. We encourage our readers to check the various video examples of generated hand trajectories included in the Supp. Mat.

### 4.4 GENERALIZABILITY TO UNSEEN OBJECTS

To test how well our model generalizes across different object instances, we evaluate on the HOI4D instance split and compare our model with the MDM-I baseline. Results in Table 2 show that our method outperforms the baseline in terms of all three metrics. By using object-centered visual features, we are able to generalize better to unseen object instances.

| Method | FPHAB *(subject)* | | | HOI4D *(location)* | | |
|---|---|---|---|---|---|---|
| | ACC ↑ | DIV → | FID ↓ | ACC ↑ | DIV → | FID ↓ |
| Real | 1.000 | 6.300 | 0.000 | 1.000 | 5.393 | 0.000 |
| T2M-T, Guo et al. (2022) | 0.124±0.011 | **6.258**±0.070 | 0.809±0.078 | 0.730±0.022 | **5.348** ± 0.179 | 1.851±0.360 |
| MDM-T, Tevet et al. (2022) | 0.284±0.016 | 6.545±0.087 | 0.818±0.058 | 0.771±0.025 | 5.680±0.280 | **1.050**±0.015 |
| MDM-I, Tevet et al. (2022) | 0.358±0.008 | 6.417±0.076 | 0.826±0.028 | 0.855±0.018 | 5.577±0.197 | 1.138±0.018 |
| Ours, obj. | 0.413±0.009 | 6.546±0.091 | **0.764**±0.001 | 0.854±0.011 | 5.454±0.151 | 1.396±0.079 |
| Ours, part | **0.417**±0.017 | 6.494±0.086 | 0.805±0.054 | **0.885**±0.011 | 5.518±0.162 | 1.436±0.164 |

Table 1: **Comparison with the state-of-the-art methods.** We compare our model SIGHT-Fusion with two other SOTA baselines and evaluate the generated hand trajectories on both FPHAB and HOI4D datasets. Note that here we use the location split of the HOI4D dataset. → indicates that values closer to "Real" are better. We observe that our method outperforms the baselines in both ACC and FID metrics on the FPHAB dataset and achieves compatible results on the HOI4D location split. The results show that image features lead to better-generated trajectories in terms of ACC and FID and the use of part features yields better outcomes compared to object features. The best performance is highlighted in **bold**, while the second-best performance is underlined.

## 4.5 DISAMBIGUATION OF ACTION BY PART FEATURES

To evaluate whether and how part features can help to disambiguate different actions and affordance, we conduct an experiment considering objects from the FPHAB dataset with multiple possible actions. These objects include *juice bottle*, *milk bottle* and *liquid soap*, and all of them are presented with the actions *open*, *close*, and *pour* in the dataset. Table 3 shows an increased accuracy (ACC) when using part features as conditioning information to the motion generator. This confirms our intuition that features from the object part help disambiguate the type of action to synthesize for multi-action objects. Note that we consider the DIV and FID metrics less relevant here due to the purpose of the experiment, yet report them for completeness.

| Cond. | HOI4D *(instance)* | | |
|---|---|---|---|
| | ACC ↑ | DIV → | FID ↓ |
| Real | 1.000 | 5.814 | 0.000 |
| MDM-I | 0.734±0.011 | 5.547±0.156 | 1.380±0.081 |
| Ours, obj. | **0.909**±0.006 | **5.780**±0.120 | **0.624**±0.050 |

| Cond. | FPHAB *(subject)* | | |
|---|---|---|---|
| | ACC ↑ | DIV → | FID ↓ |
| Real | 1.0 | 6.181 | 0.00 |
| Ours, obj. | 0.359±0.016 | **6.226**±0.147 | **2.938**±0.149 |
| Ours, part | **0.450**±0.022 | 6.064±0.114 | 3.386±0.226 |

Table 2: **Generalizability to unseen objects.** We evaluate generalization ability to unseen object instances on the HOI4D instance split and compare our model with the SOTA baseline MDM conditioned on image input. Our method outperforms the compared baseline on all metrics.

Table 3: **Disambiguation by part features.** We single out 3 objects with multiple affordances (juice bottle, milk bottle, salt), and evaluate the performance of our part $\mathcal{P}$ and object $\mathcal{O}$ models. The results show a clear increase in accuracy when using part features and indicate that actions are strongly related to object parts, and hence part features help to disambiguate the particular action.

## 4.6 EVALUATION OF PHYSICAL PLAUSIBILITY IN SIMULATOR

As an additional evaluation of the physical realism and downstream usefulness of our generated trajectories, we design environments simulating the actions found in FPHAB using the MuJoCo simulator, introduced in Todorov et al. (2012).

**Setup.** We condition our $\mathcal{P}$ model on images from the FPHAB test set, then retarget the generated human hand trajectories to a robotic hand according to Qin et al. (2021) to simulate corresponding object interactions and evaluate their success rates when executing the intended action. Specifically, we consider a total of four action-object combinations, referred to as *tasks*: *pour liquid soap*, *put salt*, *pour milk*, and *pour juice*. All evaluations of the generated trajectories start with the hand already grasping the object. The manipulated objects are initialized with several hundred particles inside. See the Supp. Mat. for details regarding the grasp initialization and trajectory retargeting. Visualizations of our environments are provided in Figure 5. Readers are further encouraged to consult the Supp. Mat. for videos of retargeted trajectories being enacted in the simulator.

**Hit rate.** For the *pour juice* and *pour milk* tasks, we simulate 280 resp. 480 spherical particles inside the opened juice/milk, and count how many particles landed inside a nearby cup out of all particles

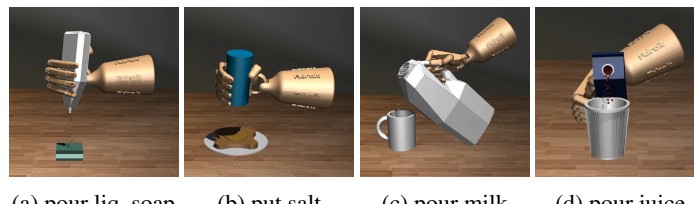

(a) pour liq. soap    (b) put salt    (c) pour milk    (d) pour juice

Figure 5: **Evaluating generated trajectories in a simulator.** We follow Qin et al. (2021) to retarget hand trajectories generated by our method to the robotic Adroit hand in the MuJoCo simulator. Afterwards, we evaluate the success rate of these trajectories in environments simulating the respective actions and objects. Four of our example environments are visualized here for the *pour liquid soap*, *put salt*, *pour milk*, and *pour juice* tasks.

that left the container while executing the retargeted trajectory. We call this metric the *hit rate*. These two tasks require the hand to rotate the juice/milk in a controlled manner, as otherwise, the liquid will immediately spill onto the floor. The receptacle (cup) slowly moves below the opening of the container (milk/juice) to catch falling particles, just as a second hand would move it during a pouring action. The *pour soap* and *put salt* tasks are similar in that a sponge resp. loaf of toast is following the container to catch emerging particles (200 for both tasks initially) whenever the salt resp. soap is shaken. Here, we again calculate the ratio of how many particles hit the receptacle out of all particles that left the container.

| Object | Juice | Milk | Salt | Soap |
|---|---|---|---|---|
| % hit, GT | 78.1 | 64.1 | 48.0 | 64.7 |
| % hit, MDM-I | 61.6 | 58.6 | 54.1 | **60.5** |
| % hit, MDM-T | 52.2 | 56.3 | **62.8** | 55.1 |
| % hit, ours | **84.4** | **65.6** | 54.6 | 60.1 |

Table 4: **Hit rates of generated trajectories.** We retarget the trajectories to a robotic hand and attempt to execute the action in MuJoCo. The favorable hit rates validate the physical realism of our generated trajectories and of our model's inference capabilities given respective features.

**Results.** We report our $\mathcal{P}$ method's hit rates for the different tasks in Table 4, and compare them to the hit rates obtained when executing retargeted ground-truth trajectories, as well as trajectories generated by the MDM-I and MDM-T baselines. Our method outperforms all settings for the *pour juice* and *pour milk* tasks and is approximately on par with the best hit rate, obtained by MDM-I, on the *pour liquid soap* task. We are only outperformed on the *pour salt* task by the MDM-T method. Note that since the FPHAB dataset contains manipulations with already empty objects, it is possible for generated trajectories to outperform the ground-truth ones due to being better aligned with the shape of the objects. When designing the objects in the simulator, we did not optimize for our method's hit rate, and instead tried to imitate the original object shape as closely as possible.

## 5 CONCLUSION

In this work, we introduce the novel task of generating natural and diverse hand trajectories conditioned on single image inputs for hand-object interaction. Our work delves into the development of methodologies for this challenging problem, presenting not only straightforward baselines, but also examining advanced approaches by proposing the novel method SIGHT-Fusion. Various hand-object interaction features are extracted from the single image input, and a diffusion-based conditional hand trajectory generative model is trained without the need for supervision by action labels. We set up comprehensive baselines and metrics on the FPHAB and HOI4D datasets, and also propose novel physical simulation environments for additional evaluation. Experiments show our superior performance over baseline methods and promising results in generalizing to unseen objects. Ablation studies further assess and validate the effectiveness of our usage of part features as conditional information during the motion synthesis. We hope that our work will invite greater interest in the SIGHT task, and that our general-purpose pipeline will be of use in related problems such as action anticipation and hand trajectory reconstruction.

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
