# OpenReview forum: "SIGHT: Single-Image Conditioned Generation of Hand Trajectories for Hand-Object Interaction"
_ICLR.cc/2025/Conference — Submitted to ICLR 2025_

### Official Review · Reviewer_ZM35 · 2024-10-30

**Soundness:** 3
**Presentation:** 2
**Contribution:** 3
**Rating:** 6
**Confidence:** 3

**Summary:**

This work propose a new task of generating hand trajctories based on a single image. And design a method called SIGHT-Fusion which is able to extract visual features and leverage diffusion-based generation model to diffuse motions. The results demonstrate well performance against baselines.

**Strengths:**

Generating hand trajectories can benefit various fields, such as VR/AR and robotics. The authors’ use of diffusion for trajectory generation is a reasonable approach to alleviate ambiguity. I also appreciate the considerable amount of video results, especially the application to dexterous hands.

**Weaknesses:**

- Most results show that the trajectory attempts to grasp a single object, which raises doubts about whether the methods have truly learned a grasp/manipulation prior. Examples such as throwing a ball or placing a pencil could better address this issue.

- Line 208 states that it focuses solely on right-hand motion, while existing hand motion generation papers, such as [1], already generate motions involving both hands.

- The architecture of SIGHT-Fusion is quite similar to MDM, while MDM achieves two-hand motions with a broader range motions.


[1] Text2HOI: Text-guided 3D Motion Generation for Hand-Object Interaction

**Questions:**

- The main question is why generating hand motions from a static image is considered valid. There could be many plausible motion trajectories that correspond to a given static image, such as moving a bottle up or down.

- What is the performance on deformable objects, such as pulling out a tissue or handling cloth?

- How can the 2.5-second SIGHT-Fusion motions be retargeted to approximately 25-second motions in the simulator? Providing more details would help readers reproduce the results.

---

> ### Author Response · Authors · 2024-11-25
>
> We thank Reviewer ZM35 for the valuable feedback and provide detailed answers to the points raised as weaknesses and questions.
>
>
> **Use of left hand**: We note that extending the method to work with both hands is mostly a limitation of the data: the FPHAB dataset features exclusively right-hand interactions, while the HOI4D dataset features right-hand interactions only for all but 2 objects. The referenced work [1] uses H2O, ARCTIC and GRAB, which record trajectories for both hands. An extension to both hands would simply consist of doubling the dimensionality of our model's output to account for another hand.
>
> **Validity of generating hand motions from a static image**: The ambiguity of human motion generation is inherent in the problem setting. In fact, the generated motions are expected to exhibit a certain diversity, which is measured by the DIV metric. The user of the model must hence expect diverse outputs if insufficient disambiguating information is provided in the input image (or text prompt in text-based motion generation methods such as [2], [3]). We further consider the study of our setting, where explicit action labels are not available and the action to synthesize has to be inferred from the image, to be a valid research question in itself.
>
> **Retargeting to simulator:** The simulator recordings in the Supp. Mat. are slowed down to allow a more detailed examination by humans. Our generated trajectories include 200 steps, and a linear interpolation of the corresponding actions is further performed to ensure a smooth execution in the simulator. Please refer to section A.6.1 in the Supp. Mat. for more details on the retargeting procedure.
>
> **Learning of manipulation prior**: The FPHAB dataset features 26 objects from a diverse range of everyday situations. Furthermore, all recordings in FPHAB start with the object already grasped, making learning the actual interaction part of the trajectory the prerequisite for good performance. We believe that our method's strong results on FPHAB provide a reasonable proof that a manipulation prior has been learned. The trajectories of the HOI4D dataset, in contrast, include the grasping of the object in addition to the manipulation itself. Here, too, our method outperforms the baselines, showing its ability to learn a useful manipulation prior.
>
> **Performance on deformable objects:**
> We extract a subset from the test set of our FPHAB subject split, and report the performance of our $\mathcal{P}$ (object+part-conditioned) and $\mathcal{O}$ (object-conditioned) models on the letter, book, paper, cloth and sponge classes of FPHAB in the table below. We further report the performance on an equally-sized subset of the remaining data, to account for the sensitivity of the FID metric to the dataset size. The diversity of the respective ground-truth data is reported for each subset.
>
> Deformable objects present a bigger challenge to identify, as their appearance may vary significantly depending on the manipulating hand's grasp. This is reflected in the lower accuracy and inferior FID compared to tests on non-deformable objects. Only in terms of diversity do we observe a better performance on the deformable subset, possibly attributable to the method learning diverse ways to manipulate a deformable object.
>
> Model | Subset | ACC $\uparrow$ | DIV $\rightarrow$ | FID $\downarrow$ |
> | --- | ---  | --- | --- | --- |
> | $\mathcal{O}$-conditioned | Deformable | $0.332 \pm 0.013$  | $\textbf{6.272} \pm 0.100 \rightarrow 6.260$ | $3.208 \pm 0.207$
> |  | Non-deformable | $\textbf{0.397} \pm 0.013$ | $6.504 \pm 0.100\rightarrow 6.310$ | $\textbf{2.347} \pm 0.179$ |
> | $\mathcal{P}$-conditioned | Deformable | $0.374 \pm 0.029$  | $\textbf{6.266} \pm 0.124 \rightarrow 6.260$ | $2.991 \pm 0.197$
> |  | Non-deformable | $\textbf{0.456} \pm 0.029$ | $6.484 \pm 0.099\rightarrow 6.310$ | $\textbf{2.356} \pm 0.177$ |
>
>
>
> We would be grateful if the reviewer would kindly consider an improved assessment of the paper. We thank you very much for your time and feedback.
>
>
>
> [1] Text2HOI: Text-guided 3D Motion Generation for Hand-Object Interaction, CVPR 2024.\
> [2] Generating Diverse and Natural 3D Human Motions From Text, CVPR 2022.\
> [3] Human Motion Diffusion Model, ICLR 2023.

---

> ### Author Response · Authors · 2024-11-29
>
> Dear Reviewer ZM35,
>
> The discussion phase is coming to an end soon, and we thus kindly request you to let us know if our response has addressed your concerns. We will be happy to answer if there are additional issues or questions, and if not, we would be grateful if you would consider a revised assessment of the review score to reflect that the issues have been addressed.
>
> Best regards,\
> Authors of SIGHT (Submission 8107)

---

> > ### Comment · Reviewer_ZM35 · 2024-12-02
> >
> > Thank you for your response, which addresses most of my concerns. I will revise my score accordingly.
> >
> > I believe that generating hand motions from static images remains a controversial topic. I acknowledge that the authors have made notable initial progress in this domain.
> >
> > This paper could be further improved if the experimental results and video demonstrations provided stronger evidence to substantiate the proposed approach. Additionally, a better justification for the choice of generating hand motions from static images would enhance the paper’s impact.

---

> ### Author Response · Authors · 2024-12-02
>
> Dear Reviewer ZM35,
>
> We thank you for your insightful review and for taking the time to address our response. We further thank you for revising your rating of our work, and are delighted that you found our response addressing your concerns satisfactory. We appreciate and have taken note of your final suggestions.
>
> Best regards,\
> Authors of SIGHT (Submission 8107)

---

### Official Review · Reviewer_5791 · 2024-11-01

**Soundness:** 2
**Presentation:** 2
**Contribution:** 2
**Rating:** 3
**Confidence:** 4

**Summary:**

They proposed a model that estimates the trajectory of the hand conditioned on a single image of hand-object interaction.

Various experiments were conducted by comparing it with models that estimate hand trajectories based on text.

**Strengths:**

They propose a model that generates the trajectory of the hand even from a single image.

The superiority of the proposed model is demonstrated through simulations.

Highly informative features are extracted by considering the contact areas between the hand and the object.

**Weaknesses:**

1. It does not consider the object's trajectory (rotation and translation), making the application of this technique seem unpractical.

2. It appears that the hand's translation is not generated separately, which also seems to be a practical issue. It seems like the fingers move and hand rotates in place.

3. The approach does not account for the interaction between the object and the hand, which makes it seem inadequate. For example, it does not consider a loss that considers actual 3D contact with objects. Also, it does not use 3D point cloud information of the object. This could prevent the network to learn generalizability to unseen objects.

4. There is no metric evaluation involving the 3D object. As HOI4D includes 3D object meshes, it seems unfortunate that there are no metrics for contact or penetration between hand the object.

**Questions:**

1. How can we confirm that the generated hand trajectory properly interacts with the object corresponding to the image? Specifically, does the part of the hand grasping the object match exactly as shown in the image? For instance, in Figure 1, the bottom of the milk carton is being grasped in the input image, but the motion generated in 3D shows that the hand is grasping the top of the milk carton.

2. Does the generated hand motion vary significantly even when the same image is used as the condition image, as shown in Figure 1?

3. What is the P model? Is it different from M? It seems to be a model not described in the Method section. From the caption of Table 3, it seems to be an other version M that uses the part features. Additionally, O is described as the interacting object in L219, but in the caption of Table 3, it is referred to as a model.

4. Is there a specific reason for not using object features and part features together?

5. Does the model perform well even in situations where there are only a few part features before averaging due to a small object in the image? In other words, I am curious if the model's performance varies significantly depending on the size of the object in the image and the number of part features.

6. Is there a reason why ours, part experiments were not conducted for unseen objects in Table 2? It would be helpful to test ours, part as well to understand hand interactions with unseen objects.

7. In the simulation examples, are the objects shown in the result images the same as those in the condition image? It would be clearer if the condition (input) images were also displayed.

---

> ### Author Response · Authors · 2024-11-26
>
> We thank Reviewer 5791 for the valuable feedback and provide detailed answers to the points raised as weaknesses and questions.
>
>
> **Absence of object's trajectory/object-related metrics:** We recognize the limitations posed by a missing reasoning about the object's trajectory. However, we believe the other contributions of our work, namely the introduction of the SIGHT task as well as a simulation-based evaluation framework for evaluating the physical plausibility of hand trajectories, together with our method's strong performance against baselines, present useful advances for the research community.
>
> **Hand's global translation:** Indeed, our method generates the global translation of the hand. In l. 255, for $J=17$, 16 joints (5 fingers x 3 joints per finger + 1 joint for the wrist) describe the pose of the hand, and one joint represents the global translation of the wrist. We merely fixed the hand's translation in the videos included in the Supplementary Material for visualization purposes.
>
> **Questions:**
>
> 1. _How can we confirm that the generated hand trajectory properly interacts with the object corresponding to the image?_\
>   Please consult Sec. 4.6, where we perform evaluations in a simulator to verify the physical plausibility of our trajectories.
>
> 2. _Does the generated hand motion vary significantly even when the same image is used as the condition image, as shown in Figure 1?_\
> The variation of generated motions is captured by the DIV metric. Please see Section 4.2 for a detailed description. As the motion sequences are initialized with random vectors before being denoised into the output sequences, there is room for variation in the final motions. Generally speaking, we find minor alternations when performing side-by-side visualizations of motions generated for the same image, verifying the stability yet diversity of the outputs of our models.
>
> 3. a) _What is the P model? Is it different from M? It seems to be a model not described in the Method section. From the caption of Table 3, it seems to be an other version M that uses the part features._\
>     The $\mathcal{P}$-conditioned model is the variation of our model conditioned on both object and part features. Please refer to Section 3.2 (l. 224) for a more detailed explanation. We would ask for the line where the "M" model is mentioned, as such a model or model variant is not introduced in our work.
>
>
>    b) _Additionally, O is described as the interacting object in L219, but in the caption of Table 3, it is referred to as a model._\
>     In l. 219, we refer to the $O$-conditioned model, i.e. the model conditioned on object features without part features.
>
> 4. _Is there a specific reason for not using object features and part features together?_\
>    Please note that part features are always used together with object features. We indicated this in line 224 with _"additional_ conditioning information", yet this may not have been clear enough. We will add a statement indicating this more explicitly in the final version of the paper.
>
> 5. _Does the model perform well even in situations where there are only a few part features before averaging due to a small object in the image? In other words, I am curious if the model's performance varies significantly depending on the size of the object in the image and the number of part features._\
> Please see our reply in the comment below.
>
> 6. _Is there a reason why ours, part experiments were not conducted for unseen objects in Table 2? It would be helpful to test ours, part as well to understand hand interactions with unseen objects._\
>   The instance split of HOI4D features images of objects by themselves, i.e. not interacted with by a hand, as described in l. 252 ff. Please consider Table 2 for results on unseen objects.
>
> 7. _In the simulation examples, are the objects shown in the result images the same as those in the condition image? It would be clearer if the condition (input) images were also displayed._\
>   The condition images for the simulation runs are 50 randomly selected images for the given action and object in the test set of the FPHAB subject split, corresponding to 50 generated trajectories to evaluate. We average over the hit rates of these trajectories to obtain the final performances in Table 4. Please consider the Supplementary Material (in `Trajectory_Visualizations/*fphab*`) for examples of the condition images for the _pour juice_ and _put salt_ actions evaluated in the simulator.
>
> We would be grateful if the reviewer would kindly consider an improved assessment of the paper. We thank you very much for your time and feedback.

---

> ### Author Response · Authors · 2024-11-26
> **Performance of P-conditioned model on images with few part patches**
>
> Our answer to question 5 was extracted to comply with comment length limits.
>
> 5. _Does the model perform well even in situations where there are only a few part features before averaging due to a small object in the image? In other words, I am curious if the model's performance varies significantly depending on the size of the object in the image and the number of part features._\
> We extract a subset from the test set of our FPHAB subject split, and report the performance of our P-conditioned (part-conditioned) model on images with less than 5 part patches (as defined in Sec. 3.2 and visualized in Fig. 2), approximately corresponding to the 10th quantile of the part patch count distribution over all samples in the test set. We further report the performance on an equally-sized subset of the remaining data, to account for the sensitivity of the FID metric to the dataset size. The diversity of the respective ground-truth data is reported for each subset.
>
> | Subset | ACC $\uparrow$ | DIV $\rightarrow$ | FID $\downarrow$ |
> | --- |  --- | --- | --- |
> | Few part patches | $\textbf{0.410} \pm 0.047$  | $6.388 \pm 0.062 \rightarrow 6.332$ | $4.930 \pm 0.613$
> | Many part patches | $0.374 \pm 0.051$ | $\textbf{6.353} \pm 0.095 \rightarrow 6.386$ | $\textbf{3.822} \pm 0.309$ |
>
> Interestingly, the accuracy of the few-patch setting is better than that of the many-patch ("Rest of dataset") one. We attribute this to a loss of disambiguating capabilities when too many patches are averaged over, some of which might not be related to the contacted object part. In terms of the diversity, the many-patch setting performs slightly better, yet both settings yield roughly comparable numbers. The FID for the few-part setting is markedly worse than that for the many-part setting. We hypothesize that averaging over more patches gives features that are more consistent across similar samples, leading to the model being able to learn to use the conditional input more effectively.

---

> ### Author Response · Authors · 2024-11-29
>
> Dear Reviewer 5791,
>
> The discussion phase is coming to an end soon, and we thus kindly request you to let us know if our response has addressed your concerns. We will be happy to answer if there are additional issues or questions, and if not, we would be grateful if you would consider a revised assessment of the review score to reflect that the issues have been addressed.
>
> Best regards,\
> Authors of SIGHT (Submission 8107)

---

> > ### Author Response · Authors · 2024-12-02
> >
> > Dear Reviewer 5791,
> >
> > As the discussion period nears its end, we would like to kindly remind you that we have addressed your concerns in our responses above, including detailed explanations and additional experiments.
> >
> > If you find our clarifications satisfactory, we would very much appreciate it if you could consider revising your rating of our work. Should you require additional clarification, please do not hesitate to reach out. We are happy to assist you until the discussion period concludes.
> >
> > Best regards,\
> > Authors of SIGHT (Submission 8107)

---

### Official Review · Reviewer_Sxj2 · 2024-11-02

**Soundness:** 1
**Presentation:** 2
**Contribution:** 1
**Rating:** 3
**Confidence:** 5

**Summary:**

This paper introduces SIGHT that generates realistic and diverse 3D hand trajectories from a single image of an object, either being interacted with or standalone. The authors propose SIGHT-Fusion, a diffusion-based conditional motion generation system that processes visual features extracted from egocentric videos of object manipulation. The method is evaluated on FPHAB and HOI4D datasets using comprehensive metrics including accuracy, diversity, and fidelity, as well as physical plausibility through MuJoCo simulation. The results demonstrate superior performance then other related works.

**Strengths:**

1. Methodology performance: The proposed SIGHT-Fusion system effectively combines advanced feature extraction techniques with a diffusion-based motion generation model, leading to improved performance over existing baselines.

2. Evaluation detail: The authors conduct extensive experiments using multiple metrics on well-established datasets and further validate their results through physics simulations. This experiment adds credibility to the physical plausibility of the generated motions.

**Weaknesses:**

The datasets used (FPHAB and HOI4D) may not include a wide variety of real-world objects and interactions, which could limit the generalization ability of the proposed method to more complex or diverse scenarios encountered in the real-world. Though authors provided the unseen testing experiment on MDM dataset, it seems also limited, compared to category-agnostic [1] or stable-diffusion-based generation method [2].
[1] HOLD: Category-agnostic 3D Reconstruction of Interacting Hands and Objects from Video, CVPR 2024.
[2] DreamFusion: Text-to-3D using 2D Diffusion, ICLR 2023.

The paper lacks an in-depth qualitative analysis of the failure cases or challenging scenarios where the proposed method might not perform well. Especially, I think it would be helpful if authors could add qualitative examples such as objects with complex shapes or scenarios with occlusion. Such analysis could provide valuable insights into the limitations and potential areas for improvement.

No discussion for previous similar literature such as [3], which is generating hand-object interactions. Though their input is different while I think the method could be trivially extends to the intended scenario.
[3] Text2HOI: Text-guided 3D Motion Generation for Hand-Object Interaction, CVPR 2024

The qualitative results do not clearly demonstrate the improvements of the proposed method. Especially, in Figure 4, a comparison between the current SOTA method and the proposed approach, it is difficult to find which result shows better quality on the figure. I request authors to have some pointers to indicate where to see.

The evaluation metrics only address limited cases. This paper addresses Hand-Object Interaction, but it does not consider any metrics to assess interaction results, such as "hand-object contact" measure. This raises concerns that the evaluation may be overly restricted to limited cases.

**Questions:**

I request authors to answer concerns raised in weakness section.

---

> ### Author Response · Authors · 2024-11-23
>
> We thank Reviewer Sxj2 for the valuable feedback and provide detailed answers to the points raised as weaknesses and questions.
>
> **Diversity of objects in FPHAB and HOI4D dataset:** FPHAB includes 26 different objects used in everyday kitchen _(fork, juice, liquid soap, milk, mug, peanut butter, salt, soda, spoon)_, office _(calculator, charger, glasses, letter, match, paper, pen, tea bag)_, and social _(card, coin, human hand itself, wallet, wine bottle, wine glass)_ scenarios, of which we train and evaluate on 25, excluding only the case where the hand itself is treated as an object in the _handshake_ and _high-five_ actions. We further train and evaluate on 9 objects _(bottle, bucket, knife, laptop, mug, pliers, scissors, stapler)_ from the HOI4D dataset.
>
> Contrary to _reconstructing_ hand-object interaction trajectories, as is the focus in HOLD [1], we do not believe it is possible to _generate_ action-appropriate hand-object interaction trajectories without at least seeing an interaction with an object of a similar object category first.
>
> We do not see a connection of our image-based hand motion generation method to text-based 3D mesh synthesis methods such as the referenced [2]. Contrary to the text-to-image setting, where ample data is available online for pretraining, there is a shortage of large-scale datasets with hand mesh data and a sufficient diversity in object categories, making it nontrivial to pretrain a hand motion generation method on large corpora that would allow it to approach category-agnostic synthesis capabilities. However, we demonstrate in Table 2 that our method is well-capable of synthesizing appropriate and realistic trajectories for images of objects from object categories seen during training.
>
> **Video comparisons with baseline:** Please consult the videos in the supplementary material for examples of trajectories generated by our method. Video comparisons of our method with the text baseline shall be included shortly.
>
> **Object-specific metrics**: Our method does not depend on/consider the object mesh when synthesizing interactions, as we focus solely on the motion of the human hand. Hence, object-specific metrics cannot be calculated.
>
> **Discussion of previous literature:** For a discussion of prior hand-motion generation literature, please consult the "Hand motion generation" paragraph of Section 2 where multiple methods are analyzed. Contrary to our method, the referenced work [3] depends on the object mesh as input. Further, it is unable to make use of datasets without 6D object pose annotations and/or action annotations, such as most of FPHAB apart from 4 kitchen objects, while our method only requires hand trajectories as ground-truth supervision during training.
>
> We would be grateful if the reviewer would kindly consider an improved assessment of the paper. We thank you very much for your time and feedback.
>
> [1] HOLD: Category-agnostic 3D Reconstruction of Interacting Hands and Objects from Video, CVPR 2024.\
> [2] DreamFusion: Text-to-3D using 2D Diffusion, ICLR 2023.\
> [3] Text2HOI: Text-guided 3D Motion Generation for Hand-Object Interaction, CVPR 2024.

---

> > ### Author Response · Authors · 2024-11-27
> > **Video comparisons with baseline**
> >
> > Dear Reviewer Sxj2,
> >
> > As requested, we provide visualizations of hand trajectories generated by our part-conditioned method using selected input images from FPHAB and HOI4D, as well as visualizations of trajectories generated by the MDM-T baseline from the same images, in https://www.dropbox.com/scl/fi/pgj14jdil98v1map357ap/SIGHT_Trajectory_Visualizations_Baseline_Comparison.zip?rlkey=7l4wl7v4s7vyrij6pkzxawd25&st=pd0l847b&dl=1. A visual comparison reveals our method's superiority in terms of both anatomical plausibility and task-appropriateness, resulting from our integration of useful visual features as conditioning information for the motion generator.
> >
> > We would be grateful if the reviewer would kindly consider an improved assessment of the paper. We thank you very much for your time and feedback.

---

> > ### Comment · Reviewer_Sxj2 · 2024-12-01
> > **Regarding object measure**
> >
> > First of all, I think authors "can" measure the object-specific metrics, by at least using the measurement like penetration ("Ptr" as in [1]) between object and hand similar to hand grasp generation method [1]. I think this aspect is important given that authors insist to capture the hand-object interactions.
> >
> > [1] Contact2Grasp: 3D Grasp Synthesis via Hand-Object Contact Constraint. IJCAI-23.
> >
> > Regarding the references, I think authors need to cite all relevant papers even though their settings are little bit different. I think mentioned things are minor and not a proper reason to omit the papers.
> >
> > I also think this is not valid enough: "we do not believe it is possible to generate action-appropriate hand-object interaction trajectories without at least seeing an interaction with an object of a similar object category first." I recommend authors to refer some of recent zero-shot interaction generation method such as "InterFusion: Text-Driven Generation of 3D Human-Object Interaction, ECCV'24". This apparently shows that something is missing in the current work.

---

> ### Author Response · Authors · 2024-11-29
>
> Dear Reviewer Sxj2,
>
> The discussion phase is coming to an end soon, and we thus kindly request you to let us know if our response has addressed your concerns. We will be happy to answer if there are additional issues or questions, and if not, we would be grateful if you would consider a revised assessment of the review score to reflect that the issues have been addressed.
>
> Best regards,\
> Authors of SIGHT (Submission 8107)

---

### Official Review · Reviewer_Yoeh · 2024-11-04

**Soundness:** 3
**Presentation:** 3
**Contribution:** 3
**Rating:** 6
**Confidence:** 5

**Summary:**

The paper introduces the task of generating 3D hand trajectories given a single image of an object, in two settings: (a) the object depicted by itself, and (b) the object being interacted with by a human hand. The authors set up comprehensive benchmarks for the new task, including various baselines and evaluation metrics. They propose a task success rate metric using physical simulation for evaluation. The paper develops SIGHT-Fusion, a novel pipeline designed to extract hand-object interaction features from a single image input and generate hand trajectories using a conditional diffusion-based model.

**Strengths:**

1. The paper proposes a challenging problem of generating plausible interaction trajectories from a single image. This problem is both interesting and important, as it addresses a critical gap in the field of 3D hand trajectory generation.
2. Using the success rate in a simulation environment as an evaluation metric is a reasonable approach, and this design is meaningful. It provides a clear and objective way to assess the performance of the proposed method.
3. The authors propose SIGHT-Fusion, which learns to generate realistic and diverse 3D hand trajectories conditioned on a single image. This is a novel and effective approach that addresses the challenges of the proposed task.

**Weaknesses:**

1. In Figure 2, when using conditions, either CLIP features or averaged features can be used. Do the authors have any experimental findings on when to use which type of condition? This design is very interesting because it can guide us on whether we should incorporate more global conditions or local conditions.
2. How does the paper handle articulated objects? Are they considered alongside rigid bodies? Articulated objects are clearly very difficult to learn with their interactive joints given an image. I noticed that HOI4D contains a large number of articulated objects, but I did not see results or visualizations regarding this aspect in the paper.
3. The paper only tests the success rate in a simulation environment on four simple tasks. I believe this evaluation metric should serve as a general metric; why not calculate the average success rate across all tasks as a quantitative measure?

**Questions:**

This paper presents a meaningful and challenging problem. However, there are still some design aspects that I am unclear about. Mainly, these concerns are related to the method of conditional injection, the handling of articulated objects, and the evaluation metrics used in the simulation environment. If the authors can address these concerns, I would be very willing to increase my score.

---

> ### Author Response · Authors · 2024-11-29
>
> We thank Reviewer Yoeh for the valuable feedback and appreciate their comments regarding the importance of the investigated problem, the meaningfulness of our simulation-based evaluation protocol, and the effectiveness of our proposed SIGHT-Fusion method. We provide detailed answers to the points raised as weaknesses and questions.
>
> **Handling of articulated objects**:  We concur with the reviewer that learning to interact with articulated objects is a difficult task, and provide experiments to support this assumption. Articulated objects are indeed found in the datasets used in our work, namely in the _open milk/juice bottle/peanut butter/liquid soap_, _close milk/juice bottle/peanut butter/liquid soap_, _flip book pages_, _unfold/clean glasses_, _open letter_ and _open wallet_ actions of FPHAB, as well as the _clamp with pliers_, _cut with scissors_, _open/close laptop_, and _use stapler_ actions of HOI4D.
>
> We evaluate our P-conditioned and O-conditioned models on a subset of the FPHAB subject split's test set including the aforementioned articulated objects, and evaluate on an equally-sized, randomly selected subset of non-articulated objects as well. An equally-sized subset is used to account for the sensitivity of the FID metric to the dataset size.
>
>
> | Experiment | Model  | ACC $\uparrow$ | DIV $\rightarrow$ | FID $\downarrow$ |
> | --- |  --- | --- | --- | --- |
> | FPHAB, Articulated  | $\mathcal{O}$-conditioned  | $0.341 \pm 0.018$  | $6.357 \pm 0.125$ $\rightarrow 6.098$ | $\textbf{2.223} \pm 0.171$  |
> |   | $\mathcal{P}$-conditioned  | $\textbf{0.380} \pm 0.027$ | $\textbf{6.174} \pm 0.121$ $\rightarrow 6.098$  | $2.368 \pm 0.211$ |
> | FPHAB, Non-articulated  | $\mathcal{O}$-conditioned | $\textbf{0.458} \pm 0.018$ | $6.538 \pm 0.114$ $\rightarrow 6.386$ | $\textbf{1.971} \pm 0.097$ |
> |   | $\mathcal{P}$-conditioned  | $0.451 \pm 0.025$ | $\textbf{6.503} \pm 0.105$ $\rightarrow 6.386$ | $2.091 \pm 0.140$ |
>
>
> We repeat the same experiment on the HOI4D dataset:
>
> | Experiment | Model  | ACC $\uparrow$ | DIV $\rightarrow$ | FID $\downarrow$ |
> | --- |  --- | --- | --- | --- |
> | HOI4D, Articulated  | $\mathcal{O}$-conditioned  | $0.870 \pm 0.011$  | $6.047 \pm 0.147$ $\rightarrow 6.154$ | $1.862 \pm 0.101$  |
> |   | $\mathcal{P}$-conditioned  | $\textbf{0.901} \pm 0.027$ | $\textbf{6.064} \pm 0.200$ $\rightarrow 6.154$  | $\textbf{0.894} \pm 0.074$ |
> | HOI4D, Non-articulated  | $\mathcal{O}$-conditioned | $0.943 \pm 0.008$ | $3.104 \pm 0.171$ $\rightarrow 3.270$ | $0.969 \pm 0.108$ |
> |   | $\mathcal{P}$-conditioned  | $\textbf{0.970} \pm 0.105$ | $\textbf{3.191} \pm 0.186$ $\rightarrow 3.270$ | $\textbf{0.622} \pm 0.112$ |
>
>
> As evident from the results, the accuracy of all models is lower on the articulated subsets of both datasets, confirming the difficulty of generating realistic interaction trajectories for articulated objects. Furthermore, the $\mathcal{P}$ (object+part-conditioned) model outperforms the accuracy of the $\mathcal{O}$ (object-conditioned) model by a remarkable margin on the articulated settings while also yielding trajectories with a better diversity, further validating the usefulness of our part-aware design.
>
> Please refer to the Supplementary Material for visualizations of trajectories generated for the _pliers_, _scissors_, _stapler_ and _laptop_ objects of HOI4D, all featuring articulation.
>
> **Details regarding conditional injection**: The conditional injection is performed by prepending object and/or part feature tokens to the input motion sequence tokens forwarded to the denoising Transformer, as explained in Sec. 3.3. The conditioning tokens are extracted as described in Sec. 3.2. One single token is used for the object and part features respectively. Please see the right side of Fig. 2 for a visualization, where we visualize a single denoising step of the diffusion model. The conditioning information is illustrated using a single blue token, while the motion sequence is illustrated using multiple red tokens.

---

> > ### Author Response · Authors · 2024-11-29
> >
> > **When to use which conditional injection**: We extend Table 3, involving the evaluation of different models' performance on multi-action objects on the FPHAB dataset, with the models' performance on an equally-sized and randomly selected subset of single-action objects. The subset is chosen to have the same size as the multi-action subset in order to account for the sensitivity of the FID metric to the dataset size. Further, we add the results on the entire FPHAB test split, i.e. all objects, from Table 1. We report the results in the table below. Note that the HOI4D dataset does not feature objects with multiple associated actions, making it impossible to conduct a similar experiment on HOI4D.
> >
> > | Experiment | Model  | ACC $\uparrow$ | DIV $\rightarrow$ | FID $\downarrow$ |
> > | --- |  --- | --- | --- | --- |
> > | Multi-action  | $\mathcal{O}$-conditioned  | $0.359 \pm 0.016$  | $\textbf{6.226} \pm 0.147$ $\rightarrow 6.181$ | $\textbf{2.938} \pm 0.149$  |
> > |   | $\mathcal{P}$-conditioned  | $\textbf{0.450} \pm 0.022$ | $6.064 \pm 0.114$ $\rightarrow 6.181$  | $3.386 \pm 0.226$ |
> > | Single-action  | $\mathcal{O}$-conditioned | $0.486 \pm 0.015$ | $6.409 \pm 0.110$ $\rightarrow 6.268$ | $2.080 \pm 0.156$ |
> > |   | $\mathcal{P}$-conditioned  | $\textbf{0.504} \pm 0.025$ | $\textbf{6.376} \pm 0.073$ $\rightarrow 6.268$ | $\textbf{2.021} \pm 0.195$ |
> > | All objects | $\mathcal{O}$-conditioned  | $0.413 \pm 0.009$ | $6.546 \pm 0.091$ $\rightarrow 6.300$ | $0.805 \pm 0.054$ |
> > | | $\mathcal{P}$-conditioned  | $\textbf{0.417} \pm 0.017$ | $\textbf{6.494} \pm 0.086$ $\rightarrow 6.300$ | $\textbf{0.764} \pm 0.001$ |
> >
> > As expected, the $\mathcal{P}$-conditioned model performs better at distinguishing which interaction to synthesize, and accordingly outperforms the object-based model in terms of accuracy in all settings. Further, it outperforms the $\mathcal{O}$-conditioned model on all metrics on the test set of single-action objects as well as the entire test set. This leads us to believe that the $\mathcal{P}$-conditioned model may be a safe choice for most settings, and the $\mathcal{O}$-conditioned model should only be used in specific cases, e.g. when more diverse trajectories are desired for multi-action objects.
> >
> >
> > **Number of tasks in simulator**: The design of environments (container and receptacle objects, success criteria) in the simulator, as well as the evaluation of methods in the simulators, is a time-consuming process, limiting the number of environments that can be included. However, we agree with the reviewer that simulation-based evaluations provide a clear and objective way of evaluting the quality and physical realism of generated motions, and that works targeting the SIGHT task should aim for a good performance in our physical simulation environments.
> >
> > **Evaluation metrics in the simulator environments**: Please consider Sec. 4.6, l. 485 ff. for a detailed elaboration of the "Hit rate" metric used to evaluate success in the simulator. The metric is based on the ratio of particles reaching a receptacle object in the designated region (e.g. inside of cup) over all objects that left the container object (e.g. juice bottle) when executing a given trajectory.
> >
> > We would be grateful if the reviewer would kindly consider an improved assessment of the paper. We thank you very much for your time and feedback.

---

> ### Author Response · Authors · 2024-11-29
>
> Dear Reviewer Yoeh,
>
> The discussion phase is coming to an end soon, and we thus kindly request you to let us know if our response has addressed your concerns. We will be happy to answer if there are additional issues or questions, and if not, we would be grateful if you would consider a revised assessment of the review score to reflect that the issues have been addressed.
>
> Best regards,\
> Authors of SIGHT (Submission 8107)

---

> > ### Author Response · Authors · 2024-12-02
> >
> > Dear Reviewer Yoeh,
> >
> > As the discussion period nears its end, we would like to kindly remind you that we have addressed your concerns in our responses above, including detailed explanations and additional experiments.
> >
> > If you find our clarifications satisfactory, we would very much appreciate it if you could consider revising your rating of our work. Should you require additional clarification, please do not hesitate to reach out. We are happy to assist you until the discussion period concludes.
> >
> > Best regards,\
> > Authors of SIGHT (Submission 8107)

---

### Meta-Review · Area_Chair_M4ss · 2024-12-21

**Metareview:**

The paper introduces a novel task of generating realistic and diverse 3D hand trajectories from a single image of an object, whether the object is standalone or involved in a hand-object interaction. The proposed SIGHT-Fusion system combines a feature extraction pipeline with a conditional diffusion-based motion generation model. Trained on video data with annotated hand trajectories without action labels, SIGHT-Fusion is evaluated on the FPHAB and HOI4D datasets using various metrics, including a new task success rate via physics simulations.

Strengths of the Paper:

-- Introduces the challenging task of generating 3D hand trajectories from a single image using a novel combination of feature extraction and diffusion-based generation.

-- Utilizes robust benchmarks with FPHAB and HOI4D datasets and includes an innovative task success rate metric through physical simulations.

-- Demonstrates superior performance over baselines, generates natural and diverse trajectories, and shows promising generalization to unseen objects.

Weaknesses:

-- Does not adequately address or visualize performance on articulated objects, despite their presence in the HOI4D dataset.

-- Lacks in-depth analysis of failure cases and interaction-specific metrics like hand-object contact measures.

-- Contains unclear methodological descriptions and insufficient differentiation from related works, such as MDM and Text2HOI.

After carefully reading the paper, the reviews and rebuttal, the AC agrees most of the reviewers that there is insufficient handling of articulated and complex object interactions, which limits its applicability to real-world scenarios. Additionally, the evaluation lacks comprehensive interaction-specific metrics, such as hand-object contact measures, undermining the thorough assessment of the generated trajectories. Thus the AC recommend to reject the paper.

**Additional Comments On Reviewer Discussion:**

See the weakness and comments above, while some of the reviews' concerns are addressed, there are still remaining concerns.

---

### Decision · Program_Chairs · 2025-01-22

Reject